# Impact of a Farm-to-School Nutrition and Gardening Intervention for Native American Families from the FRESH Study: A Randomized Wait-List Controlled Trial

**DOI:** 10.3390/nu14132601

**Published:** 2022-06-23

**Authors:** Tori Taniguchi, Alyson Haslam, Wenjie Sun, Margaret Sisk, Jann Hayman, Valarie Blue Bird Jernigan

**Affiliations:** 1Center for Indigenous Health Research and Policy, Oklahoma State University Center for Health Sciences, Tulsa, OK 74135, USA; bluebird.jernigan@okstate.edu; 2Department of Epidemiology and Biostatistics, University of California San Francisco, San Francisco, CA 94143, USA; alyson.haslam@ucsf.edu; 3Department of Rural Health, Oklahoma State University Center for Health Sciences, Tulsa, OK 74135, USA; wenjie.sun@okstate.edu; 4Osage Nation, Harvest Land, Pawhuska, OK 74056, USA; margaret.sisk@osagenation-nsn.gov (M.S.); jannhayman@osagenation-nsn.gov (J.H.)

**Keywords:** Native American, American Indian, farm-to-school intervention, early childhood education programs, community-based participatory research, randomized-controlled trial

## Abstract

Establishing healthy eating habits during childhood is critical to prevent chronic diseases that develop in adulthood. Tribally owned Early Childhood and Education (ECE) programs signify fundamental influence in childhood obesity disparities. A strategy to improve diet is the use of school gardens; however, few studies have used rigorous methods to assess diet and health outcomes. The purpose of this manuscript is to describe results from the six-month Food Resource Equity for Sustainable Health (FRESH) study among Native American families. We aimed to recruit 176 families of children attending Osage Nation ECE programs in four communities. Two communities received the intervention and two served as wait-list controls. Outcomes included change in dietary intake, body mass index, health status, systolic blood pressure (adults only), and food insecurity in children and parents. There were 193 children (*n* = 106 intervention; *n* = 87 control) and 170 adults (*n* = 93 intervention; *n* = 77 control) enrolled. Vegetable intake significantly increased in intervention children compared to controls for squash (*p* = 0.0007) and beans (*p* = 0.0002). Willingness to try scores increased for beans in intervention children (*p* = 0.049) and tomatoes in both groups (*p* = 0.01). FRESH is the first study to implement a farm-to-school intervention in rural, tribally owned ECEs. Future interventions that target healthy dietary intake among children should incorporate a comprehensive parent component in order to support healthy eating for all household members.

## 1. Introduction

In the United States, Native Americans (NAs) experience a dramatically higher burden of diet-related chronic disease across the lifespan compared to the all-race population [1]. Approximately 38% of NA adults are obese [2], and research from 2016 reports that preschool-aged NA and Alaska Native (AN) children had the highest obesity rates compared to all racial groups combined (36.7% vs. 29.1%, respectively) [3]. During childhood, establishing healthy eating habits is vital for physical growth and cognitive development [4]. Moreover, research has shown that a diet rich in vegetables during childhood can help protect against chronic diseases, such as obesity, heart disease, and diabetes, that develop during adulthood [5,6]. Though data on diet quality of NA populations are limited, prior studies that included NAs found that diet quality is insufficient and is lower than in other populations [7,8,9].

Preschool-aged children consume almost half of their daily calories at school, which is an important setting for food environment interventions [10]. Childcare-based interventions are effective in improving nutrition behaviors among children [11] and are recognized as a vital influence on learned eating behaviors [12]. However, most of these nutrition programs have been implemented in urban schools [11,13], and little is known about school-based interventions among rural NA communities. Tribally owned and operated Early Childhood and Education (ECE) programs offer preschool-aged children around two snacks and two meals per day and signify a vital organizational influence on childhood obesity disparities [14]. Therefore, ECEs can serve as an essential location to provide healthy eating interventions for NA children [14]. 

School gardens are a common strategy to increase fruit and vegetable intake in all grades [15], including ECE programs [16]; however, limited studies have used rigorous methodological designs to assess their impact on diet quality and health outcomes [14]. A systematic review on garden-based interventions among preschoolers found that only four studies assessed fruit and vegetable intake [17], and only one has been conducted among NA youth. Results from this study found that increases in preferences for vegetables were significant, but intake was not [18]. To our knowledge, there are no studies that address vegetable intake and health outcomes among NA children in ECE programs using a multi-level method, targeting the individual, family, and community. 

Using a community-based participatory research (CBPR) approach [19], we partnered with the Osage Nation to implement the Food Resource Equity and Sustainability for Health (FRESH) study, a culturally based farm-to-school intervention to increase vegetable intake among NA children and their families. The intervention was implemented within Osage Nation ECEs. The aim of this manuscript is to describe the FRESH intervention results, including changes in dietary intake (primary aim), body mass index (BMI), systolic blood pressure (adults only), health status, and food insecurity (secondary aims) among Osage Nation families.

## 2. Materials and Methods

### 2.1. Trial Design

The six-month FRESH study employed a randomized wait-list controlled trial design with treatment condition assigned at the community level (Figure 1). The design and methods of the FRESH study have been published in detail elsewhere [14]. In summary, our tribal-university partnership recruited NA families of children attending Osage Nation ECE programs in four communities to assess individual-level changes on children and adults. Two communities (*n* = 5 ECEs) received the intervention and two communities (*n* = 4 ECEs) served as wait-list controls. We randomized by community instead of ECE program to avoid crossover due to geographical proximity to members of the other study group.

### 2.2. Participant Recruitment

The FRESH Leadership Committee included four university researchers and 13 Osage Nation employees from the health, education, language, agriculture, and government divisions and led all aspects of the study. University researchers set up tables in ECE programs during school orientation, back-to-school nights, and during child drop-off/pick-up to notify parents about the study and invite them to participate [14]. ECE staff also contacted parents to notify them about the study. FRESH study flyers that promoted the study were shared through children’s backpacks and parent mailings. Flyers were also posted in classrooms around the schools. 

Adults at least 18 years old who met the following inclusion criteria were eligible to participate in the study: (1) one or more family member(s) in the household identified as NA; (2) one or more child(ren) between the ages of three and six years enrolled at an Osage Nation ECE program; (3) planned to reside in Osage Nation for nine months or more; and (4) one or more adult family member(s) willing to engage in monthly family nights at the school. Children were eligible if they were between the ages of three and six years old, enrolled in a participating ECE program, and were a household family member of an eligible adult. Adults that were interested in the study were screened and enrolled in the study if all inclusion criteria were met. Baseline data collection was then scheduled for adults and children. After baseline data collection, study parents were informed which treatment group they were in.

Written informed consent was obtained for all adults and adults gave written assent for their child(ren). All study materials were co-developed, reviewed, and approved by the Osage Nation Congress and the Oklahoma State University Center for Health Sciences Institutional Review Board.

### 2.3. FRESH Intervention Description

Three main components encompass this multi-level, randomized controlled trial: (1) a farm-to-school nutrition and garden curriculum (primary intervention); (2) an online and in-person hybrid parent curriculum (passive component); and (3) farm-to-school menu modifications at the participating ECEs.

#### 2.3.1. Farm-to-School Nutrition and Gardening Curriculum

This intervention component sought to increase vegetable intake among children in the ECE programs. We collaborated with the Leadership Committee and Osage Nation ECE program site managers, teachers, cooks, and staff at the tribal farm (Harvest Land) to determine which produce was of interest and available for the study. Since the climate in Oklahoma is unpredictable during the intervention months, ranging from cold, icy winters to a warm, wet spring, we utilized the farm and supplemented any produce they were unable to grow from a local supermarket. More details regarding the tribal farm are described elsewhere [14].

The farm-to-school nutrition and gardening curriculum was adapted for NA children from two curricula: (1) Early Sprouts [20]; and (2) Watch Me Grow [21]. More details regarding how we adapted the curriculum are described elsewhere [12]. The FRESH farm-to-school nutrition and gardening curriculum included knowledge, reading, gardening, and indoor and outdoor sensory activities, comprised of three themes that were taught for five weeks each: (1) Harvest (weeks 1–5); (2) Explore (weeks 6–10); and (3) Sprout (weeks 11–15). The focus of the curriculum was on six target vegetables: tomatoes, bell peppers, spinach, squash, butter beans, and carrots. The weekly curriculum for each theme included a reading activity (e.g., circle time or reading a book; Theme 1 only), indoor and outdoor sensory activity, and cooking activity, which included a take-home recipe kit. The intended duration for each activity varied: 5–30 min for reading activities, up to 60 min for indoor and outdoor sensory activities, and up to 75 min for cooking activities. All weekly lessons were assembled in a handbook and distributed to intervention teachers for implementation. Garden beds for the outdoor sensory activities and cooking activities were built and maintained by the Harvest Land farm staff at Osage Nation. All intervention children also took home a family recipe kit, including ingredients and a recipe to repeat the cooking activity with their family to increase exposure to the vegetables.

#### 2.3.2. Parent Curriculum

Although the FRESH study did not directly intervene upon dietary intake of parents, we did include a passive online and in-person hybrid parent curriculum, adapted from the Choose Health LA’s Healthy Parenting Workshops [22], with components from the First Nations Development Institute’s Food Sovereignty Assessment Tool [23] and the Grassroots International’s Food for Thought and Action curriculum [24]. The online curriculum comprised of 12 short video modules focused on providing parents with strategies to support their children in eating healthier foods and included healthy lifestyle education and healthy parenting practices [25]. The in-person component included three in-person family night workshops that focused on food sovereignty in the community and community capacity building for health. More details regarding the adaptation of the parent curriculum are described elsewhere [25].

#### 2.3.3. Menu Modifications

The last component included menu modifications at the ECE programs. Further description of the menu modifications from the FRESH study can be found elsewhere [26,27,28]. In short, fresh vegetables from Osage Nation’s Harvest Land farm were harvested and delivered to the ECEs to be incorporated into the ECE menus. The menus were modified to achieve best practices established by the Child and Adult Care Food Program (CACFP), which included more vegetables and fruits as snacks, replacing whole grains for refined grains, reducing fried foods, and removing sugar-sweetened beverages. The menus included the six target vegetables from the farm-to-school curriculum provided from the Osage Nation farm two times weekly and provided to the children in meals or offered as snacks within each menu cycle.

### 2.4. Trainings

Teachers in both intervention and control ECE programs received two trainings before the FRESH intervention. The first was a responsive feeding training, focusing on using “best practices” around stimulating children to try vegetables and fruits. The second training was an orientation to the FRESH farm-to-school nutrition and gardening curriculum. More information regarding teacher trainings is described in other publications [14,26,27,28].

### 2.5. Process Evaluations

We administered weekly process evaluations to assess adherence to the FRESH farm-to-school nutrition and gardening curriculum. The web-based survey was emailed weekly to teachers in the intervention ECE programs. We based the questions on the three themes and three activities (reading, sensory, and cooking) from the curriculum, which included a range of 8–10 quantitative and qualitative questions [12]. Questions included activity completion (yes or no), duration of activity (in minutes), objectives met for each activity, and a section for any comments or feedback about the weekly lesson. Teachers also indicated whether the children enjoyed, were neutral, or disliked the recipe of the week in the cooking activity section.

Fidelity was assessed by calculating the percentage of classrooms that completed each activity and was based on previous process evaluations; low fidelity (0–49%), moderate fidelity (50–74%), and high fidelity (75–100%) [29,30]. Results from the FRESH process evaluations showed high fidelity for each activity: 100% for the reading activities, 84.6–100% for the sensory activities, and 76.9–100% for the cooking activities [12].

### 2.6. Data Collection and Measures

Baseline assessments were conducted for individual-level measures on children and adults before the intervention. The intervention lasted during the spring school semester of 2018. Follow-up measures of individual-level outcomes were assessed immediately after the intervention. The primary aim was to improve dietary intake among NA children and parents. As a secondary aim, we assessed BMI, systolic blood pressure (adults only), health status, and household food insecurity among children and parents.

#### 2.6.1. Demographics

Children and adult’s age, gender, and racial/ethnic background (participants could select more than one) were assessed. In adults, we also assessed marital status, annual household income, educational attainment, employment, use of public assistance programs (e.g., Temporary Assistance for Needy Families, Supplemental Nutrition Assistance Program (SNAP), and Supplemental Security Income), number of children under the age of 18 years that live in the same household, and relationship to enrolled child(ren).

#### 2.6.2. Dietary Intake

Dietary intake for children was assessed by measuring the consumption of the six target vegetables in the FRESH farm-to-school curriculum using the weighed plate waste method [31] to assess objective levels of vegetable consumption. During the plate waste administration, we also assessed preference, or willingness to try, target vegetables. Trained researchers rated each child’s interaction with each target vegetable using a five-point checklist to measure observed willingness to try [32]. The rating options were: (0) Did not remove vegetable from box, (1) removed food, but did not bring to nose/mouth, (2) removed food and brought to nose/mouth, but did not put food in mouth, (3) put food in mouth, but did not swallow food (including taking a bite and spitting it out or licking an item), (4) put food in mouth and swallowed [32]. More information regarding the child food consumption methods is provided elsewhere [33]. 

Dietary intake for adults was evaluated using the National Cancer Institute’s Automated Self-Administered 24 h Recall [34]. Recalls were obtained either in-person or via phone by trained university staff. Recall data were used to estimate mean intake of total energy (kcals), total sugar (grams), total fats (grams), total fruits (cup equivalents), and total vegetables (cup equivalents) between intervention and control groups. We also used the 7-item Fruit and Vegetables (F/V) Behavior Checklist [35,36] to assess combined fruit and vegetable intake in cups per day.

#### 2.6.3. Biometrics and Health Status

BMI was assessed for both children and adults using measured height and weight. Weight was measured without shoes and in light clothing. Children’s height and weight were converted to BMI percentiles using the Center for Disease Control and Prevention parameters [37]. Adult height and weight were used to calculate BMI by dividing weight in kilograms by height in meters squared. Adult participants were categorized according to CDC guidelines: (1) underweight (BMI < 18.5), (2) healthy weight (BMI = 18.5–24.9), (3) overweight (BMI = 25.0–29.9), and (4) obese (BMI ≥ 30.0).

Blood pressure was measured on adults only. We used a blood pressure protocol that required adults to first empty their bladder and sit quietly for five minutes with both feet flat on the ground before blood pressure measurements. Measurements were taken three times and an average was calculated using the last two measurements. Mean systolic blood pressure is reported at baseline and after completion of the intervention.

Self-reported health status was assessed in both adults and children (by adult proxy) with five response options ranging from “excellent” to “poor”.

#### 2.6.4. Food Insecurity

To assess household food insecurity, we used the United States Department of Agriculture (USDA) 18-item Household Food Security Survey Module [38]. This survey instrument consists of 18 questions that assess quantitative and qualitative dimensions of the food supply in the household, such as behavioral and psychological responses. To calculate levels of food security, the number of confirmatory responses to the questions were totaled, counting “yes”, “often”, “sometimes”, “almost every month”, and “some months” as affirmative (score of 1). Answers consisting of “no”, “never true”, or “only 1 or 2 months” were given a score of 0. Consistent with USDA guidelines, a score of 0–1 indicates high food security, 2–3 indicates marginal food security, 3–7 indicates low food security, and 8–18 indicates very low food security.

### 2.7. Statistical Analyses

The primary outcome (dietary intake) was used to calculate the sample size. On the basis of power calculations, to detect a 15–30 g (SD = 34) difference in target vegetable consumption, the target sample size of children was 19–73 per treatment group (38–146 total). For adults, a target sample size of 168 per group (336 total) was estimated to detect a mean difference of 0.3 servings (SD = 1.2) of fruits and vegetables per day. For both calculations, 80% power and an alpha of 0.05 was set. 

To compare baseline demographic variables between intervention and control arms, a chi-square test and Fisher’s Exact test for categorical variables and an independent *t*-test for continuous variables were used. To compare difference in post-intervention outcomes between intervention and control arms, adjusted for baseline values, an analysis of covariance for continuous variables and Cochran–Mantel–Haenszel chi-squared test for categorical variables were used. All statistical analyses were performed using SAS version 9.4 [39].

## 3. Results

### 3.1. Demographics

There was a total of 193 children (*n* = 106 intervention; *n* = 87 control) in the FRESH study (Figure 1). Most (50.3%) were 4 years old; however, there was a slightly higher percentage of 3-year-olds and 5-year-olds in the control group compared to the intervention group (*p* = 0.001; Table 1). About 54.9% were female and most identified as NA/AN or white, with more intervention group children identifying as NA/AN compared to control group children (80.2% vs. 63.2%; *p* = 0.01) and more children in the control group compared to the intervention group identifying as white (59.8% vs. 27.4%; *p* < 0.001).

There was a total of 170 adults (*n* = 93 intervention; *n* = 77 control) in the study (Figure 1). The mean age was 33.2 years, the majority were female (91.8%), and over 50% reported identified as NA/AN or white, with more intervention group adults identifying as NA/AN compared to the controls (66.7% vs. 44.2%; *p* = 0.005) and more adults in the control group identifying as white compared to the intervention group (67.5% vs. 38.7%; *p* < 0.001). Almost 70% reported being married/living with a partner, 35.3% received public assistance, and the mean number of children (<18 years old) living in the household was almost three. Annual household income differed significantly between groups (*p* < 0.001), with 40.9% of intervention group participants reporting household income of more than USD 50,000 and 53.5% of control group participants reporting household income of between USD 20,000 and 50,000. Education also differed significantly between the groups (*p* = 0.02), with 32.3% of adults in the intervention group receiving a college degree or higher and 51.9% of control participants having a high school degree or less/GED. More participants in the intervention group reported working full-time compared to the control group (64.6% vs. 48.2%, respectively). Most adults reported being a parent or step-parent of the child(ren) enrolled in the study (92.4%). 

### 3.2. Child Outcomes

#### 3.2.1. Dietary Intake

Of the 193 children with baseline data, we were able to obtain follow-up data for 176 (*n* = 94 intervention; *n* = 82 control). Loss to follow-up was 8.8% among children. At baseline, plate waste results were similar between the two groups for all six vegetables. There were significant increases in vegetable intake in the intervention group compared to the controls from baseline to follow-up for squash (−0.09 to 1.3 vs. 0.3 to −0.8, respectively; *p* = 0.0007) and beans (1.4 to 3.2 vs. 1.2 to 0.7; *p* = 0.0002, respectively; Table 2). From baseline to follow-up, there were no significant differences in the intervention group compared to the control group for intake of tomatoes (5.6 to 6.7 vs. 4.4 to 5.4; *p* = 0.94), carrots (13.4 to 11.5 vs. 11.5 to 10.2; *p* = 0.76), or spinach (1.4 to 1.9 vs. 1.4 to 1.9; *p* = 0.91). The intake of peppers increased in both intervention and control groups, although the difference in intake between the groups was not significant (2.4 to 4.9 vs. 1.3 to 2.6; *p* = 0.28). 

Willingness to try results were not significantly different between the two groups at baseline for all six vegetables. There were significant increases in scores for tomatoes in both intervention and control groups (2.7 to 2.8 and 2.1 to 2.3, respectively; *p* = 0.01; Table 2) and for beans in the intervention group (2.4 to 2.7; *p* = 0.049). From the two time points, there were no significant differences in the intervention group compared to the control group for carrots (4.0 to 3.7 vs. 4.0 to 3.7; *p* = 0.5), spinach (3.3 to 3.4 vs. 2.3 to 3.1; *p* = 0.94), squash (2.0 to 2.0 vs. 1.9 to 1.9; *p* = 0.94), or peppers (2.5 to 3.1 vs. 2.0 to 2.3; *p* = 0.91). More in-depth results regarding plate weight and willingness to try data are reported elsewhere [33].

#### 3.2.2. Health Outcomes

The baseline prevalence of underweight (1.1% vs. 0%), healthy weight (67.0% vs. 65.4%), overweight (19.1% vs. 12.8%), and obese (12.8% vs. 21.8%) were not significantly different between the intervention and control groups (Table 2). Post-intervention prevalence of underweight (0.0% vs. 2.6%), healthy weight (61.8% vs. 56.4%), overweight (22.5% vs. 16.7%), and obese (15.7% vs. 24.4%) were similar in the intervention and control groups, respectively (Table 2). There were no significant changes in health status among children in either group, with most adults reporting their child had “excellent” or “very good” health at both time points.

### 3.3. Adult Outcomes

#### 3.3.1. Dietary Intake

Of the 170 adults with baseline data, we were able to obtain follow-up data for 151 (*n* = 85 intervention; *n* = 66 control). Loss to follow-up was 11.2% among adults. Fruit and vegetable intake at baseline using the F/V Behavior Checklist was similar in the intervention and control groups. Combined fruit and vegetable intake from baseline to follow-up increased slightly in the intervention group, but not in the control group (2.2 to 2.6 vs. 2.3 to 2.3, respectively), however results were not significant (*p* = 0.07; Table 3). At baseline, results from the dietary recall show there were no significant differences between the intervention and control groups for total energy, total fat, total added sugar, total fruits, or total vegetables. Results from the dietary recall show that from baseline to follow-up, the mean dietary intake of total energy and total sugar significantly improved in the intervention group (1699.9 to 1480.1; 99.0 to 75.2, respectively). Total sugar also significantly improved from baseline to follow-up among the controls (131.0 to 91.1). Total fat, total fruits, and total vegetables were not significantly different in the intervention group (68.5 to 61.2; 0.5 to 0.4; 1.3 to 1.3, respectively) or the control group (68.5 to 66.2; 0.4 to 0.5; 1.2 to 1.3, respectively) (Table 3) from baseline to follow-up.

#### 3.3.2. Health Outcomes

BMI at post-intervention was not significantly different between the intervention and control groups when adjusted for baseline values; however, most participants in both intervention and control groups were categorized as obese at baseline (54.4% and 53.0%, respectively) and post-intervention (51.4% and 53.8%, respectively). Systolic blood pressure slightly decreased in the intervention group, although not significantly different compared to the control group, from baseline to follow-up (126.5 to 120.7) (Table 3). There were no changes in health status from baseline to follow-up in either group (Table 3).

#### 3.3.3. Food Insecurity

The baseline prevalence of high food security (49.5% vs. 46.8%), marginal food security (20.4% vs. 15.6%), low food security (24.7% vs. 35.1%), and very low food security (6.5% vs. 6.5%) were not significantly different between the intervention and control groups (Table 3). After the intervention, the prevalence of high food security (58.1% vs. 62.3%), marginal food security (22.6% vs. 15.6%,), low food security (18.3% vs. 23.4%), and very low food security (2.2% vs. 2.6%) were also not significantly different between the intervention and control groups. Although more participants reported high food security in both intervention and control groups at follow-up, differences were not significant.

## 4. Discussion

The aims of the FRESH study were to improve dietary intake (primary outcome), BMI, systolic blood pressure (adults only), health status, and food insecurity (secondary outcomes) among NA families. FRESH is one of the first comprehensive multi-component, multi-level CBPR studies to use a farm-to-school and parent curricula to build community capacity and reduce obesity risk among NA families attending ECE programs [14].

Although the FRESH study did not improve BMI or other secondary outcomes among children, there were significant increases in vegetable intake. Previous studies looking at vegetable intake and BMI improvement among children showed varied results [40,41,42]. Some randomized controlled trials found that nutrition interventions that resulted in significant increase of vegetable intake also found a decrease in BMI [43], whereas others found no change among BMI [42]; the latter finding is consistent with our study. One nutrition and gardening intervention that implemented a randomized controlled trial at school found that BMI significantly improved in the intervention group compared to the controls; however, this study involved older children and a longer intervention period [44]. At follow-up, regarding the willingness to try scale, we found significant increases in scores for tomatoes for both treatment groups and increases in scores for beans in the intervention group. Our findings are similar to the *Nutrition Matters!* curriculum, which found a significant increase in willingness to try scores in three fruits and vegetables among the nutrition and gardening group [32], and is consistent with a previous study that found that repeated exposures to vegetables led to an increase in children’s willingness to try target vegetables [45].

Among the adults, the FRESH study did not improve vegetable intake, BMI, blood pressure, or food security. However, in the intervention group, there was a trend toward increased fruit and vegetable intake from baseline to post-intervention. At follow-up, total sugar intake and total energy significantly improved among the intervention group compared to the controls. Our results differ from a previous online nutritional intervention among NA participants that found an increase in vegetable intake [46]. However, as the FRESH intervention focused primarily on children with a secondary (passive) component including the parents, significant results were not expected [14]. 

Our study had several strengths. This study used a randomized controlled design, able to compare intervention and control groups. Although the study was underpowered, we still found a trend towards increased vegetable intake (*p* = 0.07) in intervention adults. In addition, we used objective measures of vegetable consumption in children rather than a dietary recall, providing a more comprehensive dietary intake. This was noted as a suggestion among authors in a systematic review on garden-based interventions among preschoolers [17]. Another strength of our study is providing children with repeat taste exposure of vegetables, which past research has shown to be effective in increasing intake [47,48,49,50,51,52,53,54,55]. Furthermore, our study focused on providing the ECE menus with local fresh vegetables, addressing the need for studies that intervene in the social determinants of health. The Osage Nation is a reservation that has limited access to healthy and fresh foods, and this study built upon and strengthened local resources by facilitating the process for supplying the ECEs with the local produce.

The limitations of this study include the challenge of implementing some of the dietary measures among children. For example, there were negative values on plate waste vegetables, which were determined to likely be the result of water condensation. In addition, we did not directly intervene with parents, which contributed to low participation rates in the study’s online component of the parent curriculum. Only 56% of parents attended the first week of the online curriculum and 12% attended the final week [25]. However, in contrast, participation in the in-person component of the parent/family intervention (e.g., family nights) was nearly twice as high as the online participation, although it also decreased as the intervention continued [25]. 

Since the original implementation of the FRESH study, the Osage Nation Harvest Land farm built upon the lessons from the study process and findings to expand its produce to a greater number of tribal programs and services. The Harvest Land farm now features commercial-grade aquaponics systems and eight new state-of-the-art greenhouses, which are able to grow various vegetables year-round. The farm is now in the process of developing a tribally specific community-supported agriculture program, which aims to increase access and intake of fresh produce.

## 5. Conclusions

The FRESH study is one of the first farm-to-school interventions, guided by a CBPR orientation, implemented in rural NA ECE programs to assess diet and health outcomes. This study adds to the limited information available with regards to vegetable intake among NA children [56]. Future interventions to encourage healthy dietary intake among young children should include a comprehensive parent component that supports healthy eating for all members of the family and household, while at the same time continuing to focus on the social determinants of health to build community capacity and systemic resilience. When intervening within NA communities, programs should incorporate use of NA languages, traditional food origin and cultural stories, and the transmission of knowledge from elders as a potential strategy for better inclusion of a family/household approach. Repeating the FRESH intervention in other NA ECE programs, with adaptations specifically tailored to their communities, will aid in testing the feasibility in diverse communities.

## Figures and Tables

**Figure 1 nutrients-14-02601-f001:**
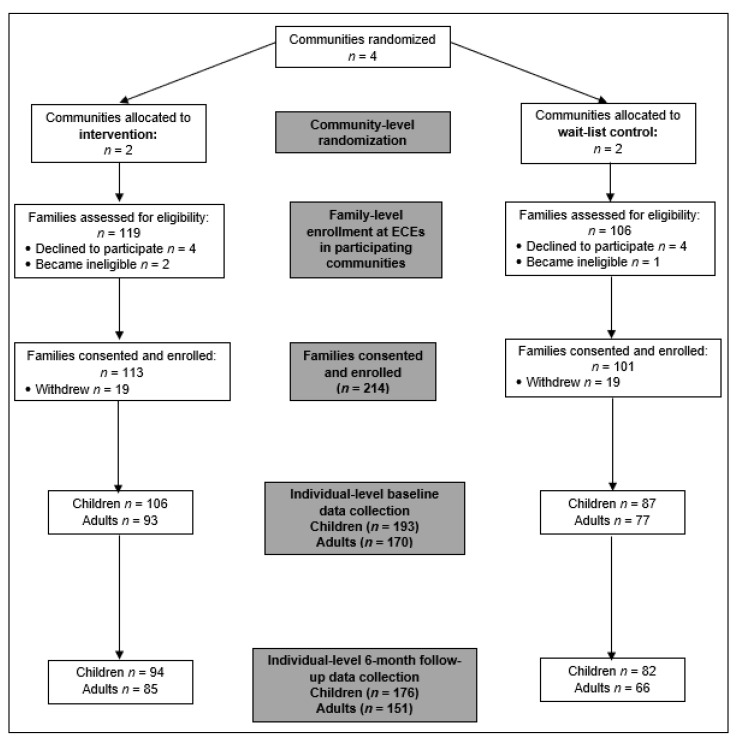
FRESH study community randomization, recruitment, and data collection in Osage Nation of Oklahoma.

**Table 1 nutrients-14-02601-t001:** Baseline characteristics for children and adults participating in the FRESH study.

Child Demographics				
	Overall *n* = 193	Intervention *n* = 106	Control *n* = 87	*p*-Value ^3^
Age, *n* (%)				**0.001**
3	42 (21.8)	18 (17.0)	24 (27.6)	
4	97 (50.3)	54 (50.9)	43 (49.4)	
5	41 (21.2)	21 (19.8)	20 (23.0)	
6 or older	13 (6.7)	13 (12.3)	0 (0.0)	
Gender, *n* (%)				0.38
Male	87 (45.1)	51 (48.1)	36 (41.4)	
Female	106 (54.9)	55 (51.9)	51 (58.6)	
Race/ethnicity ^1^, *n* (%)				
Native American or Alaska Native	140 (72.5)	85 (80.2)	55 (63.2)	**0.01**
White/Caucasian	81 (42.0)	29 (27.4)	52 (59.8)	**<0.001**
Asian or Pacific Islander	3 (1.6)	2 (1.9)	1 (1.1)	1.0
Black or African American	6 (3.1)	2 (1.9)	4 (4.6)	0.41
Hispanic/Latino	14 (7.3)	9 (8.5)	5 (5.7)	0.58
**Adult Demographics**				
	**Overall** ***n* = 170**	**Intervention** ***n* = 93**	**Control** ***n* = 77**	***p*-Value ^4^**
Age, mean (SD)	33.2 (7.1)	34.0 (7.3)	32.2 (6.8)	0.11
Gender, *n* (%)				0.64
Male	14 (8.2)	9 (9.7)	5 (6.5)	
Female	156 (91.8)	84 (90.3)	72 (93.5)	
Race/ethnicity ^1^, *n* (%)				
Native American or Alaska Native	96 (56.5)	62 (66.7)	34 (44.2)	**0.005**
White/Caucasian	88 (51.8)	36 (38.7)	52 (67.5)	**<0.001**
Asian or Pacific Islander	2 (1.2)	1 (1.1)	1 (1.3)	1.00
Black or African American	3 (1.8)	1 (1.1)	2 (2.6)	0.87
Hispanic/Latino	6 (3.5)	3 (3.2)	3 (3.9)	1.00
Married or living with partner, *n* (%)	118 (69.4)	65 (69.9)	53 (68.8)	1.00
Receive public assistance ^2^, *n* (%)	60 (35.3)	27 (29.0)	33 (42.9)	0.09
Children < 18 years living in household, *n* (%)	2.8 (1.4)	2.8 (1.4)	2.7 (1.4)	0.65
Annual household income, *n* (%)				**<0.001**
Less than $20,000	50 (29.8)	24 (25.8)	26 (34.7)	
$20,000–$50,000	71 (42.3)	31 (33.3)	40 (53.3)	
More than $50,000	47 (28.0)	38 (40.9)	9 (12.0)	
Education, *n* (%)				**0.02**
High school degree or less/GED	77 (45.3)	37 (39.8)	40 (51.9)	
Some college or technical school	52 (30.6)	26 (28.0)	26 (33.8)	
College degree or higher	41 (24.1)	30 (32.3)	11 (14.3)	
Work full-time, *n* (%)	90 (56.2)	60 (64.6)	37 (48.2)	**0.04**
Relationship to child participant, *n* (%)				0.25
Parent or step-parent	157 (92.4)	83 (88.3)	75 (94.9)	
Grandparent	10 (5.9)	8 (8.5)	3 (3.8)	
Aunt or Uncle	1 (0.6)	0 (0.0)	1 (1.3)	
Other	2 (1.2)	2 (2.1)	0 (0.0)	

^1^ Participants responded to multiple race categories. ^2^ Includes tribal government, federal government, state government (excluding casino or “oil payments”), or the state or local welfare office. ^3^ Chi-square using Fisher’s Exact test to compare characteristics between intervention and control group. ^4^ Chi-square or *t*-test to compare characteristics between intervention and control group. Bold denotes significant *p*-value < 0.05.

**Table 2 nutrients-14-02601-t002:** Differences in baseline and follow-up measurements of outcomes for children in the FRESH study.

Variable	Intervention *n* = 94		Control *n* = 82		*p*-Value ^2^
	Baseline	Follow-Up	Baseline	Follow-Up	
**Dietary Intake**					
Plate Waste, mean ± SD					
Tomatoes	5.6 ± 8.4	6.7 ± 8.9	4.4 ± 7.5	5.4 ± 9.6	0.94
Carrots	13.4 ± 12.8	11.5 ± 11.8	11.5 ± 12.8	10.2 ± 12.0	0.76
Spinach	1.4 ±1.8	1.9 ± 2.4	1.4 ± 2.0	1.9 ± 2.6	0.91
Squash	−0.09 ± 3.2	1.3 ± 4.8	0.3 ± 3.8	−0.8 ± 2.4	**0.0007**
Beans	1.4 ± 2.7	3.2 ± 5.3	1.2 ± 3.2	0.7 ± 2.0	**0.0002**
Peppers	2.4 ± 5.0	4.9 ± 8.0	1.3 ± 3.8	2.6 ± 6.9	0.28
Willingness to Try					
Tomatoes	2.7 ± 1.8	2.8 ± 1.9	2.1 ± 1.7	2.3 ± 1.8	**0.01**
Carrots	4.0 ± 1.6	4.0 ± 1.6	3.7 ± 1.8	3.7 ± 1.8	0.50
Spinach	3.3 ± 1.9	3.4 ± 1.9	2.3 ± 1.8	3.1 ± 2.0	0.94
Squash	2.0 ± 1.6	2.0 ± 1.5	1.9 ± 1.4	1.9 ± 1.4	0.94
Beans	2.4 ± 1.8	2.7 ± 1.9	2.0 ± 1.5	1.9 ± 1.5	**0.049**
Peppers	2.5 ± 1.8	3.1 ± 1.8	2.0 ± 1.4	2.3 ± 1.8	0.91
**Health Outcomes**					
BMI (kg/m^2^), *n* (%)					0.31 ^3^
Underweight	1 (1.1)	0 (0.0)	0 (0.0)	2 (2.6)	
Healthy weight	63 (67.0)	55 (61.8)	51 (65.4)	44 (56.4)	
Overweight	18 (19.1)	20 (22.5)	10 (12.8)	13 (16.7)	
Obese	12 (12.8)	14 (15.7)	17 (21.8)	19 (24.4)	
Health status ^1^, *n* (%)					0.35 ^3^
Excellent	35 (39.8)	39 (47.0)	36 (49.3)	40 (56.3)	
Very good	38 (43.2)	33 (39.8)	31 (42.5)	24 (33.8)	
Good	13 (14.8)	10 (12.0)	5 (6.8)	6 (8.5)	
Fair	2 (2.3)	1 (1.2)	1 (1.4)	1 (1.4)	
Poor	0 (0.0)	0 (0.0)	0 (0.0)	0 (0.0)	

^1^ Answered by parent proxy. ^2^ Analysis of covariance evaluating differences in post-intervention values between intervention and control groups, controlling for pre-intervention values for respective outcome variable (unadjusted for other covariates). ^3^ Cochran–Mantel–Haenszel chi-squared test *p*-value for difference between intervention arm, adjusted for intervention group and baseline values. Bold denotes significant *p*-value < 0.05.

**Table 3 nutrients-14-02601-t003:** Differences in baseline and follow-up measurements of outcomes for parents in the FRESH study.

Variable	Intervention *n* = 93		Control *n* = 77		*p*-Value ^1^
	Baseline	Follow-Up	Baseline	Follow-Up	
**Dietary Intake**					
F/V Behavior Checklist, mean ± SD (*n* = 152)					
Fruit and vegetable intake (cups per day)	2.2 ± 0.99	2.6 ± 1.10	2.3 ± 1.12	2.3 ± 1.11	0.07
Dietary recall, mean ± SD (*n* = 167)					
Total energy (kcal)	1699.9 ± 764.3	1480.1 ± 708.7 ^3^	1839.4 ± 767.2	1676.9 ± 732.1	0.14
Total fat (g)	68.5 ± 35.4	61.2 ± 37.3	68.5 ± 37.9	66.2 ± 31.6	0.38
Total added sugar (g)	17.5 ± 16.8	13.5 ± 13.4 ^3^	25.0 ± 20.0	15.5 ± 14.2 ^3^	0.79
Total fruits (cup eq.)	0.5 ± 1.0	0.4 ± 0.6	0.4 ± 0.8	0.5 ± 0.8	0.14
Total vegetables (cup eq.)	1.3 ± 1.0	1.3 ± 1.2	1.2 ± 1.0	1.3 ± 0.9	0.88
**Health Outcomes**					
BMI (kg/m^2^), *n* (%)					0.90
Underweight	1 (1.1)	1 (1.3)	0 (0.0)	0 (0.0)	
Healthy weight	20 (21.7)	18 (24.3)	20 (23.5)	16 (24.6)	
Overweight	21 (22.8)	17 (23.0)	20 (23.5)	14 (21.5)	
Obese	50 (54.4)	38 (51.4)	45 (53.0)	35 (53.8)	
Systolic blood pressure, mean ± SD (*n* = 152)	126.5 ± 16.6	123.6 ± 18.6	120.7 ± 14.4	122.3 ± 14.6	0.14
Health status, *n* (%)					0.69
Excellent	4 (4.5)	3 (3.6)	5 (6.8)	6 (8.5)	
Very good	30 (34.1)	24 (28.9)	24 (32.9)	21 (29.6)	
Good	40 (45.5)	43 (51.8)	32 (43.8)	31 (43.7)	
Fair	12 (13.6)	11 (13.3)	11 (15.1)	10 (14.1)	
Poor	2 (2.3)	2 (2.3)	1 (1.4)	1 (1.4)	
**Food Insecurity**					
Household food security, *n* (%)					0.63 ^2^
High	46 (49.5)	54 (58.1)	36 (46.8)	48 (62.3)	
Marginal	19 (20.4)	21 (22.6)	12 (15.6)	12 (15.6)	
Low	23 (24.7)	17 (18.3)	27 (35.1)	18 (23.4)	
Very low	6 (6.5)	2 (2.2)	5 (6.5)	2 (2.6)	

^1^ Analysis of covariance (continuous variables) or Cochran–Mantel–Haenszel chi-squared test (categorical variables) evaluating differences in post-intervention values between intervention and control groups, controlling for pre-intervention values for respective outcome variable (unadjusted for other covariates). ^2^ Comparison between follow-up groups was significant; baseline values were not significantly different.

## Data Availability

The data presented are accessible upon reasonable request from the corresponding author and can only be released following permission from the senior author and Osage Nation. The data are not publicly available to protect privacy of tribal citizen human subject participants.

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
