# Peer review of "Impact of a Farm-to-School Nutrition and Gardening Intervention for Native American Families from the FRESH Study: A Randomized Wait-List Controlled Trial"

_nutrients, 2022, doi:10.3390/nu14132601_

Round 1
Reviewer 1 Report
Thank you for this interesting and well written article which tackles a complex issue. The purpose, aims and outcomes of interest just need to be clearly presented, as well as interpretation and significance of the research presented.
Abstract
The abstract requires a clear aim and should describe the outcomes in more detail, e.g., dietary change in children and parents. ‘other secondary health outcomes is too vague.
1. Introduction
(36.7% vs. 29.1%) – this needs clarifying. Vx. May be inappropriate. Unclear how these figures can be compared unless we know the figures from other racial groups?
A clear aim or hypothesis is needed at the end of the introduction.
2. Materials and Methods
2.1 Trial Design, Recruitment and Enrollment
This section is a bit of a mix of ideas. I suggest the participant population be described separately from the study design, for example:
2.1 Trial Design (to describe all aspects of study design, including randomisation and Figure 1); and 2.2 Participant recruitment (to describe participant characteristics and inclusions/exclusions). Information on consent and ethics approval should also appear here.
2.2.2 Parent Curriculum
Line 135 - there is a 16-week component (is that correct?) and under Process Evaluations (Lines 232 and 245 and 369) it says 15-weeks. How many weeks was the intervention? This should be included under 2.1 Trial Design and can then be removed from other spots.
Line 142 - ‘The in-person component included three in-person family that focused on…’ This does not make sense. Is there an omitted word?
2.4.3 Biometrics and Health Status
Line 206 – insert ‘in’ before ‘light clothing’
2.4.5 Process Evaluations
This section is a mix of ideas. Both paragraphs start be referring to fidelity assessment/evaluation. I suggest you outline the purpose of the survey, describe the survey generally, i.e., web based and format of questions, then detail on what data is being captured.
Line 234 – replace ‘from’ with ‘on’
Table 1 – superscript 3 – what is being compared? Intervention vs. control groups? State this in the footnote.
3.2.1 Dietary Intake
Report percentage of participants unable to be followed up. Then, I would like to see a statement on whether baseline data were similar or different between groups.
Table 2 – superscript 2 – does this compare baseline vs. follow up or intervention vs. control? Provide more detail in footnote.
3.3.3.1 Dietary intake
First, write a statement on whether baseline data were similar or different between groups.
Table 3 – superscript 2 – what does this compare? Please add more detail in the footnote.
Superscript 3 Comparison between follow-up between groups was significant. Repeat word ‘between’ – please remove.
Total added sugar (g) has a superscript c in columns 3 and 5. What does this stand for? There is no mention of ‘c’ in the footnote.
3.3.2 Health Outcomes
Line 352 – was BP similar between groups?
3.3.3 Food Insecurity
Line 361 - You could place the percentages in brackets after each type of security (rather than string them altogether. Presently hard to read and match to outcome.
3.4 Process Evaluation – this section does not mention the word Fidelity, yet it is mentioned in Method and Discussion (reported as moderate to high fidelity). That result needs to be stated here.
4. Discussion
Starts with “The primary aim…”, however, this is first mention of primary aim. The aim/aims need to clearly described at the end of the Introduction and the Methods section needs to describe primary and secondary outcomes.
Restate the length of the intervention when describing it at start of discussion.
Discussion is brief and is largely a summary of the results, comparing to just 2 references from the literature. The purpose of the discussion is to interpret the results and describe the significance of the findings within the context of current literature and gaps in knowledge and to provide insights into these results. Please amend and refer to relevant literature.
Line 407 – delete ‘on’
Line 421 – rewrite sentence, suggest: ‘which were determined to be the likely result of water condensation’
Line 423 – insert full stop after ‘curriculum’
5. Conclusion
This is not just a conclusion on the results of this study. It contains contextual information that would read better as part of the discussion. The authors might mention here how this new knowledge contributes to gaps in research and can be used in practice and/or what further research is needed to support the purpose and outcomes of the research and should be similar to abstract conclusion.
Author Response
Response to Reviewer 1 Comments
Abstract
Point 1: The abstract requires a clear aim and should describe the outcomes in more detail, e.g., dietary change in children and parents. other secondary health outcomes is too vague.
Response 1: We thank the reviewer for this comment. We have added more clarity to the outcomes evaluated by listing out each outcome in the abstract.
Introduction
Point 2: (36.7% vs. 29.1%) – this needs clarifying. Vx. May be inappropriate. Unclear how these figures can be compared unless we know the figures from other racial groups?
Response 2: This is a good point by the reviewer. We have clarified the comparison, by stating that the 36.7% and 29.1% refer to NA/Alaska Native and all racial groups combined.
Point 3: A clear aim or hypothesis is needed at the end of the introduction.
Response 3: We have added a clear aim at the end of the introduction stating the intent of the manuscript.
Materials and Methods
Trial Design, Recruitment and Enrollment
Point 4: This section is a bit of a mix of ideas. I suggest the participant population be described separately from the study design, for example: 2.1 Trial Design (to describe all aspects of study design, including randomization and Figure 1); and 2.2 Participant recruitment (to describe participant characteristics and inclusions/exclusions). Information on consent and ethics approval should also appear here.
Response 4: We thank the reviewer for this feedback. We have changed the manuscript to reflect the above suggestion.
Parent Curriculum
Point 5: Line 135 - there is a 16-week component (is that correct?) and under Process Evaluations (Lines 232 and 245 and 369) it says 15-weeks. How many weeks was the intervention? This should be included under 2.1 Trial Design and can then be removed from other spots.
Response 5: The entire study was 6 months long. The farm-to-school curriculum was 15 weeks in length (with process evaluations) and the parent curriculum was 16 weeks in length. We see how stating the different lengths of each component is confusing and have edited accordingly.
Point 6: Line 142 - ‘The in-person component included three in-person family that focused on…’ This does not make sense. Is there an omitted word?
Response 6: We thank the reviewer for catching this oversight. We have added the missing words, clarifying that it was family night workshops.
Biometrics and Health Status
Point 7: Line 206 – insert ‘in’ before ‘light clothing’
Response 7: “In” was added before “light clothing”.
Process Evaluations
Point 8: This section is a mix of ideas. Both paragraphs start be referring to fidelity assessment/evaluation. I suggest you outline the purpose of the survey, describe the survey generally, i.e., web based and format of questions, then detail on what data is being captured.
Response 8: We agree with the reviewer. The process evaluation results have been previously reported on in another manuscript so we changed this paragraph to briefly describe the methods and results for the process evaluations and included citations.
Point 9: Line 234 – replace ‘from’ with ‘on’
Response 9: I replaced “from” with “on”.
Point 10: Table 1 – superscript 3 – what is being compared? Intervention vs. control groups? State this in the footnote.
Response 10: I added the comparison group in the footnote on superscript 3 in table 1. You are correct in that the test was comparing intervention and control groups.
Dietary Intake
Point 11: Report percentage of participants unable to be followed up. Then, I would like to see a statement on whether baseline data were similar or different between groups.
Response 11: We have added the loss to follow-up percentage. We also added a statement on whether baseline was similar or different between the groups. We thank the reviewer for this feedback.
Point 12: Table 2 – superscript 2 – does this compare baseline vs. follow up or intervention vs. control? Provide more detail in footnote.
Response 12: We thank the reviewer for this comment and agree that the methods in the footnote were unclear. We have added clarification to superscript 2.
Dietary intake
Point 13: First, write a statement on whether baseline data were similar or different between groups.
Response 13: We have added the loss to follow-up percentage for adults and we also added a statement on whether baseline was similar or different between the groups.
Point 14: Table 3 – superscript 2 – what does this compare? Please add more detail in the footnote.
Response 14: Again, we appreciate the opportunity to provide more clarity to the footnote.
Point 15: Superscript 3 Comparison between follow-up between groups was significant. Repeat word ‘between’ – please remove.
Response 15: The extra word has been removed.
Point 16: Total added sugar (g) has a superscript c in columns 3 and 5. What does this stand for? There is no mention of ‘c’ in the footnote.
Response 16: The superscript in the table is supposed to say “3” not “c”. Thank you for catching this. I have changed the table to reflect what is in the footnote.
Health Outcomes
Point 17: Line 352 – was BP similar between groups?
Response 17: We are not able to state that the two groups are similar – only that we did not find a difference. We have clarified the sentence to indicate that post-intervention differences were not different.
Food Insecurity
Point 18: Line 361 - You could place the percentages in brackets after each type of security (rather than string them altogether. Presently hard to read and match to outcome.
Response 18: I agree. I added the percentages in parenthesis after each type of security for both baseline and follow-up and re-worded the sentences to make sense.
Process Evaluation
Point 19: This section does not mention the word Fidelity, yet it is mentioned in Method and Discussion (reported as moderate to high fidelity). That result needs to be stated here.
Response 19: This has already been reported on (Wetherill MS, Bourque EE, Taniguchi T, Love CV, Sisk M, Jernigan VB. Development of a Tribally-led Gardening Curriculum for Indigenous Preschool Children: The FRESH Study. Journal of Nutrition Education and Behavior. 2021 Nov 1;53(11):991-5) so we defined fidelity, summarized the results, and included citations in the methods section.
Discussion
Point 20: Starts with “The primary aim…”, however, this is first mention of primary aim. The aim/aims need to clearly described at the end of the Introduction and the Methods section needs to describe primary and secondary outcomes.
Response 20: Thank you for this comment and we agree that the aims needed clarity at the end of the introduction, methods, and discussion sections. We have edited the manuscript in these sections to provide more description on what the primary and secondary aims were.
Point 21: Restate the length of the intervention when describing it at start of discussion.
Response 21: I added the length of the intervention in line 387.
Point 22: Discussion is brief and is largely a summary of the results, comparing to just 2 references from the literature. The purpose of the discussion is to interpret the results and describe the significance of the findings within the context of current literature and gaps in knowledge and to provide insights into these results. Please amend and refer to relevant literature.
Response 22: We thank the reviewer for the feedback. We have lengthened the discussion and have also added findings from other previous research to compare/contrast findings from our study.
Point 23: Line 407 – delete ‘on’
Response 23: “On” is now deleted.
Point 24: Line 421 – rewrite sentence, suggest: ‘which were determined to be the likely result of water condensation’
Response 24: Thank you for this suggestion. I changed the sentence to reflect your revision.
Point 25: Line 423 – insert full stop after ‘curriculum’
Response 25: I added a period after “curriculum”.
Conclusion
Point 26: This is not just a conclusion on the results of this study. It contains contextual information that would read better as part of the discussion. The authors might mention here how this new knowledge contributes to gaps in research and can be used in practice and/or what further research is needed to support the purpose and outcomes of the research and should be similar to abstract conclusion.
Response 26: We thank the reviewer for this helpful feedback. We agree that the conclusion should be similar to the abstract conclusion and we have added how this study contributes to research gaps. We also moved the last paragraph from the Strengths and Limitations and added it to the conclusion since it contains information about what is needed for future research.
Reviewer 2 Report
The manuscript is of interest to Native Americans and other native minorities in other countries. In addition to value, the paper also has some problems.
1. As a non-American, I am unfamiliar with Oklahoma's climate and agriculture. Some basic information should be appreciated
2. For the purpose of this review, I myself analyzed Oklahoma's climate and agriculture, and found some discrepancies between the background data and the information contained in the article. The study concluded that the main goal of this study was to increase the access and consumption of fresh vegetables among NA children. The intervention lasted for five months in the 2018 spring semester (January to May). I think Oklahoma's climate is not conducive to the cultivation of fresh vegetables at least in the first quarter of the year. This should be clearly explained.
3. Why was blood pressure only measured in adults? Many children suffer from high blood pressure, especially obese children. This should be an important aspect of this work - dietary modification of the components of the metabolic syndrome. If not for laboratory parameters, blood pressure should at least be measured in children, since it was measured in adults.
4. On the other hand, there is a stratification of BMI in children, and why there is no stratification of BMI in adults, after all, most of the participants were classified as obese.
5. The discussion is short - in my opinion, it does not fully exhaust the topic and the research carried out
6. Lack of software manufacturers.
Author Response
Response to Reviewer 2 Comments
Point 1: As a non-American, I am unfamiliar with Oklahoma's climate and agriculture. Some basic information should be appreciated
Response 1: We appreciate this feedback. We added some description to the climate during the intervention months in Section 2.2.1.
Point 2: For the purpose of this review, I myself analyzed Oklahoma's climate and agriculture, and found some discrepancies between the background data and the information contained in the article. The study concluded that the main goal of this study was to increase the access and consumption of fresh vegetables among NA children. The intervention lasted for five months in the 2018 spring semester (January to May). I think Oklahoma's climate is not conducive to the cultivation of fresh vegetables at least in the first quarter of the year. This should be clearly explained.
Response 2: Thank you for looking into the background on the climate and agriculture in Oklahoma. We have changed the manuscript to clarify that the main goal of the study was to increase dietary intake, not access. The reviewer is correct, that the climate in Oklahoma is unpredictable for growing and harvesting produce in these months. Along with the addition in Response 1 above, the authors added that the farm was used for the produce they could grow and that the rest of the produce needed for the intervention was supplemented by a local supermarket.
Point 3: Why was blood pressure only measured in adults? Many children suffer from high blood pressure, especially obese children. This should be an important aspect of this work - dietary modification of the components of the metabolic syndrome. If not for laboratory parameters, blood pressure should at least be measured in children, since it was measured in adults.
Response 3: We agree that measuring blood pressure in children is important, however Osage Nation did not feel comfortable having us conduct blood pressure measurements in children so we only collected height and weight in terms of biometrics for them. I added this as a limitation to the study in the Discussion section (line XX).
Point 4: On the other hand, there is a stratification of BMI in children, and why there is no stratification of BMI in adults, after all, most of the participants were classified as obese.
Response 4: We have changed the BMI results to show categorical instead of continuous data to match the child BMI outcomes. Indeed, most participants were obese and we agree that stratifying by categories is better to show the percentage of participants who fall in the obese category.
Point 5: The discussion is short - in my opinion, it does not fully exhaust the topic and the research carried out.
Response 5: We thank the reviewer for the feedback. We have lengthened to discussion section by adding more information regarding the research. We also included findings from other similar studies to compare results.
Point 6: Lack of software manufacturers.
Response 6: We have added a citation to the SAS software used for analysis, but we are unsure what this comment is exactly referring to. We would be happy to add this information if the reviewer could be more specific in what they are asking.